# Cross Pseudo Method for Semi-supervised Abdominal Organ Segmentation

Junhao Dong[1], Tianyu Xiao[1], Ruining Zhao[1], Ziqi He[1], and Xu Ji[1]

[1]School of Artificial Intelligence, Beijing University of Posts and Telecommunications, Beijing 100876, China

**Abstract.** Abdominal multi-organ segmentation is of great significant for preoperative treatment planning. At present, there are many public abdominal datasets and deep learning based segmentatiomethods have been proposed. However, the problem of polycentric and spatio-temporal inefficiency still remain unsolved. Meanwhile, expensive costs of labeling and lack of labeled data are also serious problems of this field. In this work, with a small amount of labeled CT images and large number of unlabeled data, we propose a novel Cross Pseudo based semi-supervision method, whose two branches can generate pseudo-labels to supervise each other. For quantitative evaluation on the FLARE2022 validation cases, this method achieves the DSC of 0.80, NSD of 0.75 within merely 20s for inference per image.It demonstrates the robustness and generalization of our method.

**Keywords:** Multi-organ Segmentation · Cross Pseudo Supervision · Semi-supervised Learning.

## 1 Introduction

The emergence of more and more difficult diseases in the world has become the driving force for the rapid development of the medical field. The invention of medical imaging techniques, such as X-ray imaging, computed tomography (CT) and magnetic resonance imaging (MR), has greatly promoted clinical work, especially the study of abdominal organs.

Medical image semantic segmentation is an important tool in clinical practice. It is used for accurately delineating tumors and treating certain cancers in radiotherapy, and for morphological analysis of organs to infer information such as the volume and shape of liver. It is well known that the segmentation of human organs on CT images is a very difficult task. Artificial organ contour rendering not only has a large number of internal and inter-observer differences, but also needs to bear the risk of complex organ contour and prone to pathological changes. Therefore, it is not practical to draw the contour of the organ by hand. This has led to an increased demand for automated methods for medical image segmentation. Accurate semantic segmentation of abdominal organs in clinical research has become a key research topic in the field of intelligent medicine in recent years. Nowadays, many semantic segmentation methods[1][3][5][8] based on

deep learning have been proposed, and they have achieved great success in heart image segmentation and brain tumor segmentation. Many of the methods are based on the well-known U-Net[13][15] architecture, which works well on many datasets. However, these methods still have the following three disadvantages at present.

First of all, most methods are primarily aimed at maximizing the accuracy of predictions without considering the efficiency of models, which often leads to increased complexity of models and thus reduces their applicability in clinical practice. Second, many organ segmentation data contain only images from a single center or scanner, or only cases of a single disease, so the generalization properties of models trained on these data are not always very good. Finally, most of the existing well-performing semantic segmentation monitoring methods rely on large scale annotated data[6][16]. However, due to the particularity of medical images, the cost of obtaining a large number of marked data is too high to be realistic. In order to avoid the dilemma of large amount of labeled data, a semi supervised semantic segmentation method was proposed, which proposed the idea of learning models from a few labeled images and a large number of unlabeled images.

In recent years, two main methods of semi-supervised learning, namely, consistent regularization[19] and entropy minimization[9], have been proposed. Consistent regularization facilitates the model to produce stable and consistent predictions for the same unlabeled data under various perturbations, such as shape and color. On the other hand. Entropy minimization uses unlabeled data in an explicit bootstrap manner, that is, assigning false labels to unlabeled data and training them jointly with manual label data. Unlike previous work, MixMatch[2] leverages the best of both approaches and presents a hybrid framework that leverages unlabeled data from both perspectives. FixMatch[18] inherits the spirit of MixMatch, but simplifies unnecessary mechanics.

Semi-supervised semantic segmentation tend to utilize the Generative Adversarial Networks[7] (GANs) as an auxiliary supervision signal for the unlabeled data. However,GANs are not easy to optimize and may suffer the problem of mode collapse.As an extension of FixMatch, PseudoSeg[21] adapts the weak-to-strong consistency to segmentation scenario and further applies a calibration module to refine the pseudo masks. Based on the above research, we propose a novel cross pseudo supervision method, which has two branches to generate pseudo labels to supervise each other. Our contributions are as follows:

- We propose a novel semi-supervised method for Abdominal Organ Segmentation, which uses cross-pseudo-label supervision.

- We leverage a combination of cross-entropy loss and the cross pseudo supervision loss, which is verified compatible and efficient.

- Experiments show the robustness and generalization of our method.

## 2    Method

In this section, we propose a novel cross pseudo supervision based method for abdominal organ segmentation with the SOTA segmentation network deeplabv3+. A detail description of the method is as follows.

### 2.1    Preprocessing

Our proposed method includes the following preprocessing strategies:

- **Reorientation image to target direction.** According to the format of SimpleITK, the directions of the z-axis, y-axis, and x-axis are respectively adjusted to [1 -1 -1].
- **Convert 3D image to 2D.** We unroll the 3D image along the Z axis, turning it into slices.
- **Data Augmentation.** First, the image is clipped to the range [-100, 1000] after organ-intensity jitter. Then a random rotation is applied followed by z-score normalization based on the mean and standard deviation of the intensity values. Besides, Cutmix[20] is also employed.

### 2.2    Proposed Method

The proposed cross pseudo method consists of two branches with the same architecture. Each branch generates pseudo-labels to supervise the other branch, thereby improving the robustness and generalization of the model. The experiment uses the current state-of-the-art segmentation network deeplabv3+.

**Network Architecture.** Figure 1 illustrates the utilized segmentation network Deeplabv3+ [4]. When it comes to the backbone network, we select ResNet18[10] with relatively small amount of parameters to prevent overfitting. Note that the provided annotations are too limited to be used for deeper architecture, and the competition prohibits any pretrained weights. DeepLabv3+ also use dilated convolutions and atrous spatial pyramid pooling (ASPP) to obtain multi-scale contextual features, as shown in Figure 2, which can capture organ features of different sizes. DeepLabv3+ also employs a decoder module that concatenates low-level high-resolution features and high-level semantic features to refine segmentation boundaries. Specifically, the high-level semantic features generated by ASPP are 4× upsampled, while the shallow features pass through a 1×1 convolutional layer before combination. Then the final prediction can be obtained by a 3×3 convolutional layer and 4× upsampling.

**Semi-Supervised Strategies.** We propose a novel semi-supervised method based on cross pseudo supervision (CPS)[5] to leverage unlabeled data and optimize the parameters. The main idea is illustrated in Figure 3.

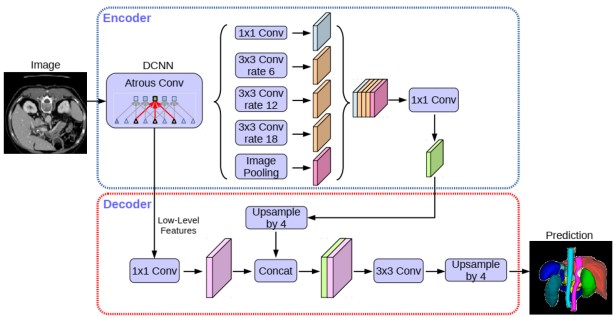

**Fig. 1.** Illustration of the network architecture.

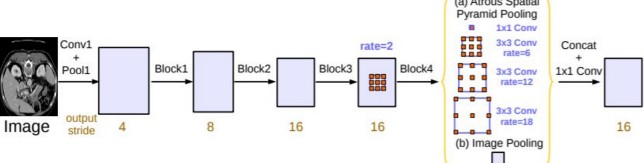

**Fig. 2.** Dilated convolution and ASPP.

We have two branches of networks with the same architecture initialized variously. First, the same labeled and unlabeled images are fed into these two networks respectively, and the segmentation results $P$ can be obtained, which represent the predicted probability of each class per pixel. Then, we achieve the pseudo labels $Y$ with a Softmax layer for each branch. For labeled data, we calculate the supervision loss for each branch, and for unlabeled ones, a cross pseudo supervision loss is calculated. Multi-branch networks and cross pseudo supervision boost the generalization and robustness of the whole model.

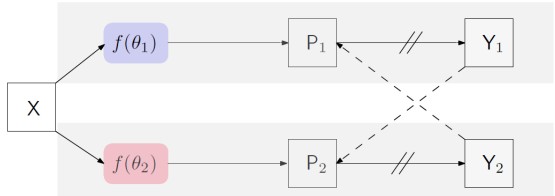

**Fig. 3.** Main idea of CPS.

**Loss function.** For supervision loss, we use the pixel-wise cross-entropy loss formulated as follows:

$$\mathcal{L}_s = \frac{1}{|\mathcal{D}^l|} \sum_{\mathbf{X} \in \mathcal{D}^l} \frac{1}{W \times H} \sum_{i=0}^{W \times H} \left( \ell_{ce} \left( \mathbf{p}_{1i}, \mathbf{y}_{1i}^* \right) + \ell_{ce} \left( \mathbf{p}_{2i}, \mathbf{y}_{2i}^* \right) \right) \tag{1}$$

where $\ell_{ce}$ is the cross-entropy loss function and $\mathbf{y}_{1i}^*$ $(\mathbf{y}_{2i}^*)$ denote the ground-truth label for supervision. $H$ and $W$ are the height and width of the input images.

The cross pseudo supervision loss for unlabeled data is defined as:

$$\mathcal{L}_{cps}^u = \frac{1}{|\mathcal{D}^u|} \sum_{\mathbf{X} \in \mathcal{D}^u} \frac{1}{W \times H} \sum_{i=0}^{W \times H} \left( \ell_{ce} \left( \mathbf{p}_{1i}, \mathbf{y}_{2i} \right) + \ell_{ce} \left( \mathbf{p}_{2i}, \mathbf{y}_{1i} \right) \right) \tag{2}$$

where $\mathbf{p}_{1i}$ and $\mathbf{p}_{2i}$ are the outputs of the networks, and $\mathbf{y}_{2i}$ and $\mathbf{y}_{1i}$ represent the pseudo labels to supervise bidirectionally as shown in Fig. 3. The whole loss function can be formulated as:

$$\mathcal{L} = \mathcal{L}_s + \lambda \mathcal{L}_{cps} \tag{3}$$

In this paper, we set $\lambda$ as 0.5.

### 2.3 Post-processing

We add a bias of 0.6 for each class except background before Softmax, which verifies an improvement in performance. Areas with less than 50 pixels are also considered as noise and filtered out.

## 3 Experiments

### 3.1 Dataset and evaluation measures

The FLARE2022 dataset is curated from more than 20 medical groups under the license permission, including MSD [17], KiTS [11,12], AbdomenCT-1K [14], and TCIA [6]. The training set includes 50 labelled CT scans with pancreas disease and 2000 unlabelled CT scans with liver, kidney, spleen, or pancreas diseases. The validation set includes 50 CT scans with liver, kidney, spleen, or pancreas diseases. The testing set includes 200 CT scans where 100 cases has liver, kidney, spleen, or pancreas diseases and the other 100 cases has uterine corpus endometrial, urothelial bladder, stomach, sarcomas, or ovarian diseases. All the CT scans only have image information and the center information is not available.

The evaluation measures consist of two accuracy measures: Dice Similarity Coefficient (DSC) and Normalized Surface Dice (NSD), and three running efficiency measures: running time, area under GPU memory-time curve, and area under CPU utilization-time curve. All measures will be used to compute the ranking. Moreover, the GPU memory consumption has a 2 GB tolerance.

### 3.2   Implementation details

Not that we use ResNet-18 as the backbone of DeepLabv3+ instead of deeper architectures to alleviate overfitting as well as improve inference speed and reduce resource consumption. Other details are illustrated as follows.

**Environment settings**  The environments and requirements are presented in Table 1.

**Table 1.** Environments and requirements.

| | |
|---|---|
| Windows/Ubuntu version | Ubuntu 18.04.4 LTS |
| CPU | Intel(R) Xeon(R) CPU E5-2698 v4 @ 2.20GHz |
| RAM | 503GB |
| GPU (number and type) | NVIDIA Tesla V100 32G (×8) |
| CUDA version | 10.2 |
| Programming language | Python 3.7 |
| Deep learning framework | Pytorch (Torch 1.8.0, torchvision 0.9.0) |

**Training protocols**  The training protocols of the proposed method is shown in Table 2.

**Table 2.** Training protocols.

| | |
|---|---|
| Network initialization | Kaiming normal initialization |
| Data augmentation methods | Cutmix, Clip, Intensity Jitter, Random Rotation |
| Patch sampling strategy | Slice 3D images along the Z axis |
| Batch size | 96 |
| Patch size | 512×512 |
| Total epochs | 400 |
| Optimizer | SGD with momentum 0.1, weight decay: 0.0001 |
| Loss | Cross Entropy Loss, Cross Pseudo Supervision Loss |
| Initial learning rate (lr) | 0.02 |
| Lr strategy | WarmUpPoly |
| Training time | 12 hours |

## 4   Results and discussion

### 4.1   Quantitative results on validation set

Table 3 illustrates the validation performance (DSC) comparisons between with and without using unlabeled images. It can be seen that there is an obvious

improvement with using unlabeled data, both for each organ and the mean value. It demonstrates the effectiveness of our method utilizing a large number of unlabeled examples.

**Table 3.** DSC comparisons between with and without using unlabelled images. Ours(w/ un): with unlabeled data; Ours(w/o un): without unlabeled data

| Organ | Ours(w/ un) | Ours(w/o un) |
|---|---|---|
| Liver | 0.9559 | 0.9375 |
| RK | 0.9008 | 0.8296 |
| Spleen | 0.9208 | 0.8784 |
| Pancreas | 0.7981 | 0.6586 |
| Aorta | 0.9459 | 0.9193 |
| IVC | 0.8544 | 0.7809 |
| RAG | 0.6302 | 0.5138 |
| LAG | 0.6023 | 0.4116 |
| Gallbladder | 0.6134 | 0.4643 |
| Esophagus | 0.745 | 0.6815 |
| Stomach | 0.7886 | 0.7879 |
| Duodenum | 0.6781 | 0.5448 |
| LK | 0.9118 | 0.8426 |
| Average | **0.7958** | **0.7116** |

### 4.2   Visualized examples of successful and failed cases

Fig. 4 demonstrates the visualized examples of successful and failed cases. It can be seen that our model still suffers false positive problems,i.e., the background tends to be predicted as organs.

### 4.3   Segmentation efficiency results

The average running time is 20.34s per case in validation set, and Maximum used GPU memory is 1917 MB, which means a full score in this evaluation. Besides, we obtain an area under GPU memory-time curve and an area under CPU utilization-time curve of 31508.74 and 405.96.

### 4.4   Limitations and future work

Our model tends to predict the background pixels as the pixels of the organs, which significantly degrades the accuracy. Future work will focus on proposing a suitable post-processing technique and an advanced training strategy.

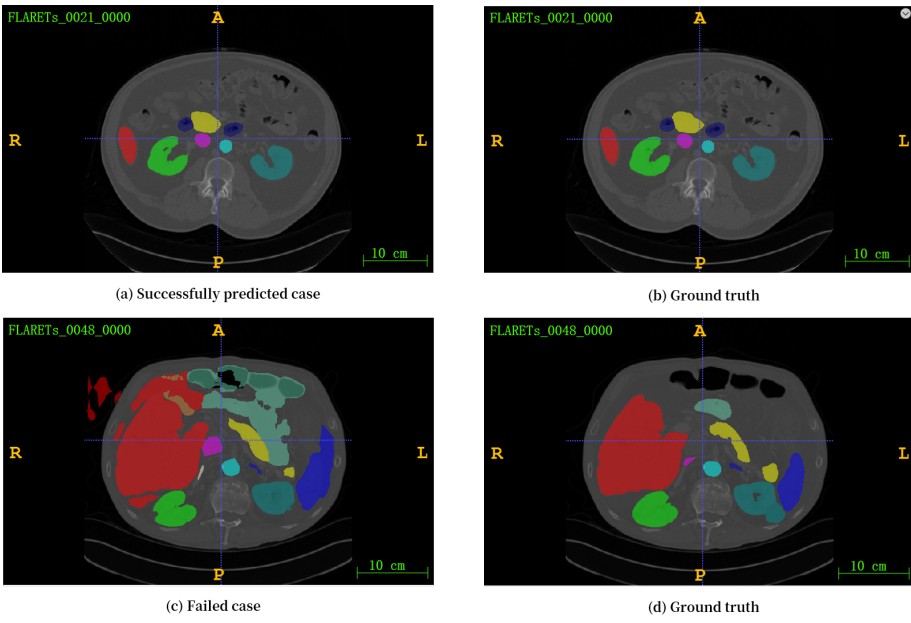

(a) Successfully predicted case
(b) Ground truth

(c) Failed case
(d) Ground truth

**Fig. 4.** Visualized examples of successful and failed cases.

## 5   Conclusion

In this work, we propose a novel Cross Pseudo based semi-supervision method, whose two branches can generate pseudo-labels to supervise bidirectionally. Experiments have verified that our model can leverage a large amount of unlabeled CT images and greatly improve the performance. Futher work will be performed on an appropriate post-processing technique and an advanced training strategy.

**Acknowledgements** The authors of this paper declare that the segmentation method they implemented for participation in the FLARE 2022 challenge has not used any pre-trained models nor additional datasets other than those provided by the organizers, and the proposed solution is fully automatic without any manual intervention.

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
