# OpenReview forum: "Cross Pseudo Method for Semi-supervised Abdominal Organ Segmentation"
_MICCAI.org/2022/Challenge/FLARE_

### Official Review · Reviewer_pBTA · 2022-09-12
**The paper provides sufficient and clear details of their work**

**Rating:** 6
**Confidence:** 4

**Review:**

Authors apply recent state-of-the-arts Cross Pseudo Supervision (CPS) with DeeplabV3+ network. There is no significant modification in the method but they gave detailed equation, explanation, configuration and experiment in this report of theirs.

Some suggested improvement:
- More qualitative results and comparison
- Figure 3 is cropped from the original paper, it should be redrawn
- Table 3 compares between using and not using unlabeled data. What model did you use to train/evaluate in the “without unlabeled data" setting in this comparison?
- How do you do inference ? By ensembling both deeplabv3+ models ?
- Why didn't you use the traditional Dice loss

---

### Official Review · Reviewer_TJya · 2022-09-16
**Do not copy figures and formulas directly from other works.**

**Rating:** 3
**Confidence:** 5

**Review:**

**Suggest improvements:**

- Do not copy figures and formulas directly from other works.
- No ablation experiment for the proposed post-processing method:

​		The proposed method is a 2D model, and the method segments organs slice by slice. So for some specific slices, it's common that the area of pixels of small organs or other edge slices of the organ will be less than 50 pixels. It's unreasonable to filter out an area under 50 pixels. Will these areas be filtered out like noise? Is sure that this post-processing method will improve performance?

---

### Official Review · Reviewer_PiNh · 2022-09-18
**Typo issue and DSC needs improved.**

**Rating:** 5
**Confidence:** 4

**Review:**

Strengths: The proposed method achieves semi-supervised abdominal organ via cross pseudo supervision with a mean DSC of 0.80, NSD of 0.75 within 20s for inference. The descriptions of the method and the analysis are clear.
Weakness:
The core idea of cross pseudo supervision comes from existing work. The authors need to highlight their contributions.
Some typos in the abstract.
The DSC of 0.80 still needs to be improved.

---

### Official Review · Reviewer_aL84 · 2022-09-19
**Cross pseudo is simple and effective**

**Rating:** 6
**Confidence:** 4

**Review:**

1. This work proposed a semi-supervised method for Abdominal Organ Segmentation using cross pseudo. While this approach already seems to have some similar work, the idea is simple and effective.
2. Table 3 has shown that the model with cross pseudo training can outperform the one without a lot, demonstrating the proposed method's effectiveness. However, the improvement in the segmentation performance of the stomach is almost non-existent. Does this mean that the method is not very effective for such an organ that will squirm and is irregularly shaped, and is it possible to design some modules to solve these challenges?
3. By the way, there are some minor mistakes, such as misspellings and no partitions of the word ("segmentatiomethods" in the abstract) and missing spaces after the period ("image.It" in the abstract).

---

### Official Review · Reviewer_xb9V · 2022-09-20
**The inference time of the model is good, and the paper is well written but lacks novelty.**

**Rating:** 6
**Confidence:** 4

**Review:**

1. The authors may not claim CPS as their own contribution in the article.
2. The results of this model could be compared with that of common baselines in terms of accuracy and efficiency.

---

### Official Review · Reviewer_95Xp · 2022-09-20
**Cross Pseudo Method for Semi-supervised Abdominal Organ Segmentation**

**Rating:** 6
**Confidence:** 3

**Review:**

Strengths: The proposed method achieved efficient and effective semi-supervised learning with a mean DSC of 0.7958 and a mean inference time of 20 s. The proposed method used dilated convolution and ASPP to build a lightweight network and applied cross pseudo supervision as the semi-supervised method.

Weaknesses:
* Although semi-supervised learning effectively improved the DSC from the baseline 0.7116 to 0.7958, the final score is still not as high as that of other papers. This may be attributed to using 2D inputs rather than 3D? The reason for using 2D inputs rather than 3D is not clear.
* The bias of 0.6 for the foreground classes used in post-processing needs more explanation, and there seems to be a risk of being overfit to the validation set.

---

### Meta-Review · Program_Chairs · 2022-09-28

**Recommendation:** Major Revision
**Confidence:** 5

**Metareview:**

Reviewers raise many concerns and suggestions. Please address all comments in the revised manuscript.